# A Method to Evaluate Spectral Analysis by Spectroscopy

**DOI:** 10.3390/s22155638

**Published:** 2022-07-28

**Authors:** Yan Liu, Pingping Fan, Huimin Qiu, Xueying Li, Guangli Hou

**Affiliations:** 1Institute of Oceanographic Instrumentation, Qilu University of Technology, Qingdao 266061, China; sdqdliuyan@126.com (Y.L.); qiuhm@qlu.edu.cn (H.Q.); ponneylxy@163.com (X.L.); hgl@sdioi.com (G.H.); 2Shandong Key Lab of Marine Environment Monitoring Technology, Qingdao 266061, China

**Keywords:** detection resolution, organic carbon, sediment, spectrometer, spectral analysis

## Abstract

Visible and near infrared spectroscopy has been widely used to develop a method for rapidly determining organic carbon in soils or sediments (SOC). Most of these studies concentrated on how to establish a good spectral model but ignored how to evaluate the method, such as the use of detection range (max and min), resolution and error for SOC spectral analysis. Here, we proposed a method to evaluate the spectral analysis of SOC. Using 96 sediments sampled in the Yellow Sea and Bohai Sea, China, we established three spectral models of SOC after collecting their spectral reflectance by Agilent Cary 5000, ASD FieldSpec 4 and Ocean Optics QEPro, respectively. For both the calibration set and validation set in each spectrometer, the predicted SOC concentrations followed a distribution curve (function), in which the *x*-axis was the SOC concentrations. Using these curves, we developed these four technical parameters. The detection ranges were the SOC concentrations where the curve was near to or crossing with the lateral axis, while the detection resolution was the average difference between the two neighboring SOC concentrations. The detection errors were the differences between the predicted SOC and the measured SOC. Results showed that these technical parameters were better in the bench-top spectrometer (Cary 5000) than those in the portable spectrometers when analyzing the same samples. For the portable spectrometers, QEPro had a broader detection range and more consistent detection error than FieldSpec 4, suggesting that the low-cost QEPro performed as well as the high-cost FieldSpec 4. This study provides a good example for evaluating spectral analysis by spectroscopy, which can support the development of the spectral method.

## 1. Introduction

Visible and near infrared reflectance (Vis-NIR) spectroscopy has been widely used to rapidly analyze many properties in various types of soils or sediments [1,2,3,4,5,6,7,8,9,10,11,12]. As the primary chemical attribute, organic carbon in soils or sediments (SOC) is directly related with Vis-NIR reflectance and has been mainly investigated for 20 years [13]. Past studies showed that more effort was concentrated on how to establish an accurate spectral model of SOC [1,3,13]. Therefore, many studies paid attention to the use of varied algorithms in the whole process of spectral analysis to accurately predict SOC using Vis-NIR spectroscopy. For example, Liu et al. (2019) studied the spectral analysis of SOC in Chinese forests at the large (country) scale and found that the use of cubist after clustering could greatly improve the accuracy of SOC prediction compared with partial least squares regression (PLSR) [3]. At the local scale, PLSR was usually used to establish spectral models but showed distinctive results in different studies [13].

Obviously, spectrometers play an important role in spectral analysis [10,14,15,16,17,18,19]. During the past three years, many studies have investigated the influences of spectrometers on the spectral analysis of SOC (Table 1). Results showed that different spectrometers could produce varying spectral models, with distinctive R^2^, RMSE and RPD or RPIQ, even for the same samples (Table 1). For example, SOC in the Rothamsted long-term field experiments was analyzed by ASD FieldSpec 3 and Ocean Optics STS, respectively [15]. The spectral model derived from FieldSpec 3 had higher R^2^ and RPD, and lower RMSE than that from STS, and the difference became smaller outdoors compared with the laboratory (Table 1) [15]. The results also showed that laboratory spectrometers had a better performance than the field ones due to their technical advantages in terms of signal-to-noise ratio, stability of the measurement conditions, spectral resolution and wide spectral range [15,17,19]. For example, as mid-IR spectrometers, Vertex 70 (Bruker Optics, Ettlingen, Germany) and Agilent 4200 (Agilent Technologies, Santa Clara, CA, USA) are bench-top and portable, respectively [20]. SOC in an Australian farm was analyzed by Vertex 70 and Agilent 4200, respectively. The results showed that the spectral model derived from Vertex 70 was better than that from Agilent 4200 (Table 1) [20].

However, spectrometers in the comparison studies had distinct classifications, belonging to near infrared and mid infrared instruments (Table 1) [14,17,19]. As shown in Table 1, Agilent 4200, Agilent 4300 (Agilent Technologies, Santa Clara, CA, USA) and Bruker are all mid infrared spectrometers (mid-IR); the others are near infrared spectrometers (near-IR). The same samples produced different reflectance spectra from the mid-IR spectrometers compared with near-IR. Therefore, because the spectrometers were different in type, the comparison could not really be shown as to which spectrometer was better in their performance on spectral analysis.

Studies also showed that ASD spectrometers are the most commonly used in the Vis-NIR spectral analysis due to their good performance and reputation [6,7,8,9,10,11,12,13,14,15,17,18,19]. However, ASD spectrometers are expensive, being a barrier to studies on Vis-NIR spectroscopy [13,18]. Fortunately, there are many other successful spectral analysis studies using other Vis-NIR spectrometers [14,15,16,17,18,19]. Crucil et al. (2019) analyzed SOC in the Rothamsted long-term field experiments by Ocean Optics STS, showing a good performance on spectral analysis [15]. Li et al. (2021) analyzed the total carbon of the intertidal sediments in the Laizhou Bay of China using QE65000 (Ocean Optics, USA) and achieved good prediction results, with r^2^ > 0.87 and RPD = 2.8 [4].

These studies suggested the importance of selecting a suitable spectrometer for a certain specific research [10]. However, how to evaluate the performance of a spectrometer on spectral analysis is not clear. Evaluating a spectral model usually utilizes three parameters, including the determination coefficient (R^2^), the mean root square error (RMSE) and the residual predictive deviation (RPD) in the validation set [20,21]. A good spectral model usually has high R^2^ and RPD, and low RMSE in the validation set [20,21]. If the RPD is higher than 2.0, the spectral model is considered to be excellent and can be used for accurate quantification [20]. This evaluation can only resolve whether the spectral model is appropriate for spectral analysis, but can not resolve for what samples the spectral analysis is suitable, such as the basic technical parameters, including the detection range, error, and resolution.

Eskildsen et al. (2016) proposed a method to evaluate spectral analysis across spectrometers [22]. Different from the old method comparing the difference between predicted and measured values, this method compared the median of the predicted values to evaluate spectral analysis, showing that this method was more useful than the old method [22]. These results suggested the importance of exploring the predicted values when evaluating spectral analysis [22].

Inspired by Eskildsen et al. (2016) [22], we made full use of the predicted SOC in spectral models to evaluate the spectral analysis of SOC in the Yellow Sea and Bohai Sea, China. We also compared the performance of three common near-IR spectrometers on the spectral analysis of SOC, including Agilent Cary 5000 (bench-top), ASD FieldSpec 4 and Ocean Optics QEPro. Suggested by the past studies (Table 1) [14,17,19], we hypothesized that the performance on spectral analysis would be ordered as Cary 5000 > FieldSpec 4 > QEPro.

## 2. Materials and Methods

### 2.1. Sampling and Preparation

Ninety-six sediments were sampled in the stations of the Yellow Sea and Bohai Sea, China. For detailed sampling sites see Qiu et al. 2022 [23]. After freeze drying, the sediments were ground and sieved to pass a 0.5 mm mesh. Each sample was divided into two parts, one for chemical determination and the other for collecting the Vis-NIR reflectance spectra.

SOC concentrations were measured using the Vario EL III Elemental Analyzer after the inorganic carbon was removed by hydrochloric acid. Specifically, 1 g of dried sample mixed with 1 mL hydrochloric acid (1 mol/L) was ultrasonic for 3 h to remove inorganic carbon, which was repeated 2 times. Organic carbon concentrations in these sediments ranged from 0.25% to 1.73%, with an average of 0.75% and a median of 0.67%.

### 2.2. Reflectance Spectra Collection

The reflectance spectra were collected using Cary 5000 (Agilent Technology, Santa Clara, CA, USA), FieldSpec 4 (Analytical Spectral Devices, Boulder, CO, USA) and QEPro (Ocean Optics, Dunedin, FL, USA). The specifications of these spectrometers were shown in Table 2. Cary 5000 is a bench-top lab spectrometer, thus has the best specification. Both FieldSpec 4 and QEPro are portable and can be used both in the lab and field. The three spectrometers had different specifications with the price ordered as QEPro < Cary 5000 < FieldSpec 4. The wavelength range of both Cary 5000 and FieldSpec4 is 350–2500 nm, while that of QEPro is 200–1000 nm. QEPro had the largest stray light, without the detail of wavelength repeatability and accuracy.

Spectra collection was controlled by the software of each spectrometer. Cary 5000 has a special diffuse reflectance module for powder samples (DRA-2500, Agilent), which can be used to collect the reflectance spectra of soils or sediments. Different from Cary 5000, both FieldSpec 4 and QEPro are fiber spectrometers, which collect reflectance spectra by optical fibers. In QEPro, the optical fiber was inserted into the 45° hole of the probe bracket (RPH-1, Ocean Optics). The sample box was under the probe bracket and formed a 45° angle between samples and the probe. The vertical distance between the probe and the sample was 0.5 mm.

Some parameters were set up by software. In Cary 5000, the wavelength interval was 1 nm, the scanning speed was 600 nm/min, and 5 replicates were set for each sample. In FieldSpec 4, 3 replicates were set each time, with 100 times of dark current, 10 times of reference spectrum and 5 times per sample to collect the reflectance spectra, and the white board was used for calibration every 15 min while collecting the spectra. In QEPro with the slit of 10 μm, the integration time was 600 ms, the sampling interval was 1 nm and each sample had 5 replicates of reflectance spectra.

### 2.3. Spectral Analysis

Each sample had a spectral curve, whose *x*-axis was wavelength (nm) and *y*-axis was reflectance (%), and each sample had its SOC concentrations. Therefore, 96 samples had 96 curves and corresponding SOC concentrations. Using these data, the relationship (spectral model) between reflectance spectra (many x) and SOC concentrations (y) could be established. Specifically, 96 samples were divided into a calibration set and a validation set by Kennard–Stone algorithm (K-S) at the ratio of 2:1 [24]. Each sample had many x (reflectance), i.e., x1, x2, x3, … xn were the reflectances at each wavelength, and one y (SOC concentration). Qiu et al. (2022) reported that the spectral model established by the whole spectra had much better prediction ability of total carbon and nitrogen concentrations, maybe because SOC concentrations were reflected by all the wavelengths [23]. Therefore, the spectral model was also established by the whole spectra without characteristic spectra extraction in this study. Confalonieri et al. (2001) reported that partial least squares regression (PLSR) was successful in modelling soil nutrients by near infrared spectroscopy [25], as many other studies found [13,25]. PLSR was also found to be perfect for total carbon and nitrogen modelling in the same sediments, especially using the whole spectra [25]. Therefore, in this study PLSR was used to establish the spectral model of SOC. Lastly, the spectral model was evaluated by the validation set using the determination coefficient (R^2^), the mean root square error (RMSE) and the residual predictive deviation (RPD) [20,21].

$\mathrm{RPD}=\mathrm{SD}/\mathrm{RMSE}=\sqrt{\frac{1}{n-1}\sum_{i=1}^{n}{\left(y_{i}-\bar{y_{i}}\right)}^{2}}/\sqrt{\frac{1}{n-1}\sum_{i-1}^{n}(y_{i}-\overset{\wedge }{y_{i}}{)}^{2}}$, in which $y_{i}$ is the measured value of the sample, $\hat{y_{i}}$ is the predicted value by the model, $\bar{y_{i}}$ is the average measured value of the sample and n is the number of samples. Rossel et al. (2006) proposed a quantitative relationship between RPD and model quality: RPD > 2.0, the models or predictions are perfect and can be used for rapid analysis [20].

All analysis was performed in MATLAB 2017b. Specific programs had been published in the Chinese software copyright (2018SR325853). A new program in R is being prepared.

### 2.4. Evaluation of the Spectral Analysis

Four parameters were proposed to evaluate the spectral analysis from different spectrometers, including the detection resolution, range (minimum and maximum) and error. These parameters were calculated according to the distribution of predicted SOC concentrations (Figure 1).

These four parameters were calculated as shown in Figure 2. The distributions of predicted SOC concentrations were fitted as a function for both calibration sets and validation sets, whose *x*-axis was SOC concentrations. Under the condition that the function is near to or equals a constant (e.g., 0), the x-values (SOC concentrations) are the minimum and maximum (detection range). The detection resolution is the average difference between the two neighboring SOC concentrations. The detection errors are the difference between the predicted SOC and the measured SOC, which can be plotted by the quartile map. The quartile map can directly show the actual error including the largest error, the smallest error and the median error.

All analysis was performed in SPSS 13.0 (SPSS Inc., Cary, NC, USA); a new program in R is being prepared. Here, we stress the procedure of how to calculate detection range. In SPSS, the predicted SOC was first analyzed by “Graphs”-“Histogram”. Then, the distribution of predicted SOC was performed by “Analyze”-“Regression”-”Nonlinear”-“Peak-Gaussian” or “Analyze”-“Fit curve”-”Cubic”. Based on the function, the extreme could be calculated.

## 3. Results

### 3.1. Reflectance Spectra of the Same Sediments in Different Spectrometers

The reflectance spectra collected by the three spectrometers were different (Figure 3 and Figure 4). As shown in Figure 3, QEPro only collected the reflectance spectra below 1000 nm, while both Cary 5000 and FieldSpec 4 collected the complete visible and near infrared reflectance spectrum (350–2500 nm). At 500–1000 nm, the reflectance was mainly ranged at 20–50% in Cary 5000, 15–45% in FieldSpec 4 and 20–60% in QEPro. All spectral curves in Cary 5000 were more compact than the portable spectrometers.

As shown in Figure 4, the differences among these three spectrometers were direct and simple. The shapes of the spectral curves were the same, especially for Cary 5000 and FieldSpec 4. The reflectance was ordered as QEPro > Cary 5000 > FieldSpec 4.

### 3.2. Spectral Analysis from Different Spectrometers

The spectral analysis was highly different across spectrometers. In the calibration sets, Cary 5000 had the best model with the largest R^2^ and the smallest RMSE, while QEPro had the worst model with the smallest R^2^ and the largest RMSE (Figure 4). In the validation sets, Cary 5000 also had the best model with the largest R^2^, the smallest RMSE and the highest RPD, while FieldSpec 4 had the worst model with the smallest R^2^, the largest RMSE and the lowest RPD (Figure 5). These results showed that Cary 5000 had the best performance on the spectral analysis of SOC.

### 3.3. Evaluation of the Spectral Analysis

Firstly, the errors of spectral models both in the calibration set and the validation set were the smallest from Cary 5000 but the largest from QEPro (Figure 6). These results were consistent with results of the spectral analysis (Figure 5). For the portable spectrometers, the errors in FieldSpec 4 increased from the calibration set to the validation set, but errors decreased in QEPro (Figure 6), showing that QEPro had a consistent performance on spectral analysis.

Secondly, the spectral curves were close to a normal distribution, except for the calibration set of FieldSpec 4 (Figure 7). Cary 5000 had a wider detection range of SOC than the portable spectrometers (Figure 7). The calibration set in QEPro had a wider detection range than the validation set, showing a consistent detection range of SOC (Figure 7). These results were consistent with those in Table 3.

Thirdly, the detection resolutions were all less than 0.01% for the three spectrometers, and were not significantly different across spectrometers (Table 3). All spectrometers increased their resolution from the calibration set to the validation set, especially QEPro (Table 3).

Clearly, Cary 5000 could predict a larger range of SOC concentrations with the smallest error and highest resolution (Table 3). FieldSpec 4 had better performance in the calibration set but lost the advantage in the validation set (Table 3). Although the technical parameters of spectral analysis in QEPro had no advantages, they were consistent in both the calibration set and validation set (Figure 7, Table 2).

## 4. Discussion

We studied the spectral analysis of SOC in the Yellow Sea and Bohai Sea of China using Agilent Cary 5000, ASD FieldSpec 4 and Ocean Optics QEPro. We proposed four parameters to evaluate the spectral analysis of SOC, including the detection range (max and min), resolution and error. These parameters were used to evaluate the performance of the spectrometers on spectral analysis.

### 4.1. Advantages of the Proposed Method on Evaluation of Spectral Analysis by Spectrometers

Results showed a large difference in the spectral analysis among the three spectrometers. Consistent with our hypothesis, the portable spectrometers had the worse spectral analysis than the bench-top spectrometer (Figure 6 and Figure 7, Table 3). This was mainly due to the large difference in specification, including spectral resolution, system signal-to-noise ratio, stray light, detector, wavelength repeatability, wavelength accuracy and so on (Table 2) [15,16,19]. In this study, Cary 5000 had better specifications than the other two portable spectrometers, especially in terms of the signal-to-noise ratio, stray light, wavelength repeatability and wavelength accuracy (Table 2). Although it was easily hypothesized, we showed the quantitative differences between the spectrometers, making the evaluation quantitatively.

These results were consistent with the past studies (Table 1) [15,16,19]. Sharififar et al. (2019) analyzed SOC on an Australian farm using Brucker Optik Vertex 70 (bench-top) and Agilent 4200 (portable) and found that the spectral model derived from Vertex 70 was better than that from Agilent 4200 (Table 1) [19]. Linderhom et al. (2019) used ASD LabSpec and two other portable spectrometers to classify different soils and found that the lab spectrometer had a better classification ability than the portable spectrometers, mainly because the lab spectrometer had a wider spectral range and a better detector [26]. Unfortunately, these were qualitative studied, so we could not compare their performance on the spectral analysis by the technical parameters shown in Table 3.

This study also provided a scientific basis for evaluating spectral analysis. Although the combination of R^2^, RMSE and RPD can evaluate whether a spectral analysis is better, it is difficult to quantify how much better it is. Using the proposed parameters, we could clearly understand which samples would be suitable for the spectral technique we established, such as its detection range, resolution and error. That is, we could accurately define samples by the concentrations that could be determined (detection range) and differentiated (detection resolution) at what errors. In this study, if the SOC concentration of a sediment sampled from the Yellow Sea and Bohai Sea of China is about 2.15%, it would be more accurately analyzed by Cary 5000. If the SOC concentration is about 2.00% and the detection error is not concerning, QEPro would be more suitable.

Furthermore, if we develop a technique or an equipment that can accurately and rapidly analyze SOC based on vis-NIR spectroscopy, we should present its technical parameters, in which the proposed four parameters are the most basic. Therefore, the proposed four parameters are valuable for the equipment or a technique based on Vis-NIR spectroscopy. Spectral models are highly dependent on modelling samples; therefore, further study should pay attention to the accurate and rapid simulation of the distribution curve of the predicted SOC concentrations.

### 4.2. Performance of the Portable Spectrometers on Spectral Analysis

Inconsistent with our hypothesis, the portable spectrometers had a similar performance on spectral analysis. The detection range, error and resolution in FieldSpec 4 were not significantly different from those in QEPro (Figure 6 and Figure 7, Table 3), but QEPro had a more consistent detection range (Figure 7, Table 3) and errors (Figure 6, Table 3) in both the calibration set and the validation set than FieldSpec 4. This suggested that QEPro has a more stable performance than FieldSpec 4, at least in this study.

Yahaya et al. (2015) found that the spectral model established by the QE65000 spectrometer could be transferred to the FieldSpec 3 spectrometer more easily and in a better way than another portable spectrometer, suggesting that the two spectrometers could be replaced by each other [27]. In this study, QEPro and FieldSpec 4 were the upgraded spectrometers of QE65000 and FieldSpec 3, respectively, suggesting that QEPro and FieldSpec 4 also could replace each other.

These results indicate that if the error is not the main consideration, QEPro is a good choice due to its high stability and low price, suitable for long-term and in situ (online) use. However, if we want to establish a rapid spectral measurement technology to replace the traditional lab chemical analysis, Cary 5000 is undoubtedly a better choice. This study provided a direct reference for selecting different spectrometers to conduct effective scientific research.

## 5. Conclusions

We studied the spectral analysis of organic carbon in sediments by three spectrometers and proposed four parameters to evaluate the performance of these spectrometers. The results showed that the bench-top spectrometer had better performance on spectral analysis than the portable spectrometers, mainly due to their differences in specification, such as signal-to-noise ratio, stray light, detector, wavelength repeatability, wavelength accuracy and so on. For the two portable spectrometers, FieldSpec 4 had the best performance in the calibration set, but lost its advantage in the validation set. QEPro had ordinary technical parameters on spectral analysis, but was consistent in both the calibration set and validation set. These results suggest that if the stability is the main consideration, QEPro is a good choice for long-term and in situ use due to its low price. This study provides a direct reference for selecting different spectrometers to conduct effective scientific research, and also provides a scientific basis for evaluating the spectral analysis.

## 6. Patents

Chinese invention: A method for obtaining the performance of a spectrometer (CN202011288620.1).

## Figures and Tables

**Figure 1 sensors-22-05638-f001:**
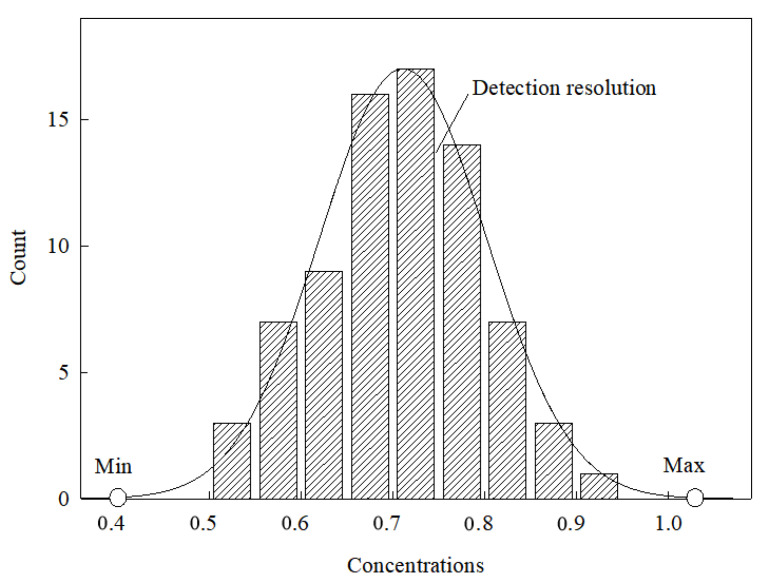
The concept map of the parameters for evaluating spectral models.

**Figure 2 sensors-22-05638-f002:**
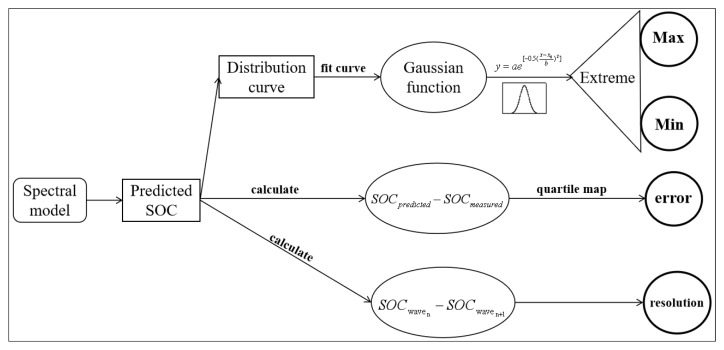
The flow chart for calculating the proposed parameters.

**Figure 3 sensors-22-05638-f003:**
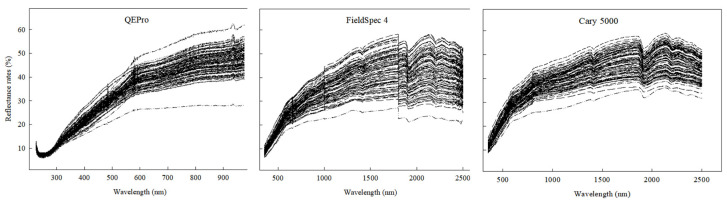
The reflectance spectra of sediments in Huanghai and Bohai Sea of China by different spectrometers.

**Figure 4 sensors-22-05638-f004:**
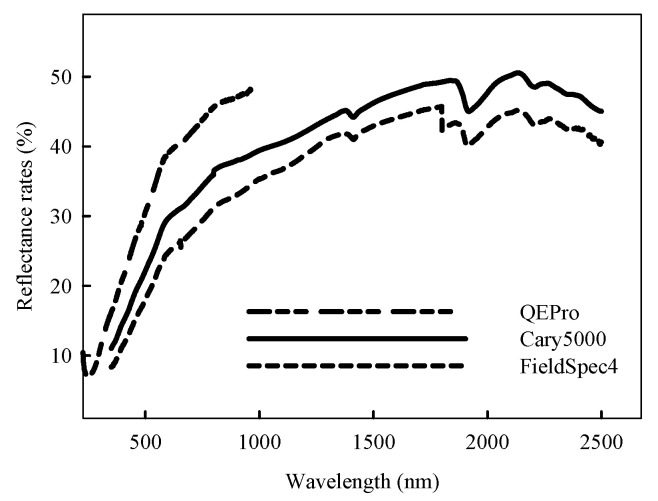
The average reflectance spectra of sediments in Huanghai and Bohai Sea of China by different spectrometers.

**Figure 5 sensors-22-05638-f005:**
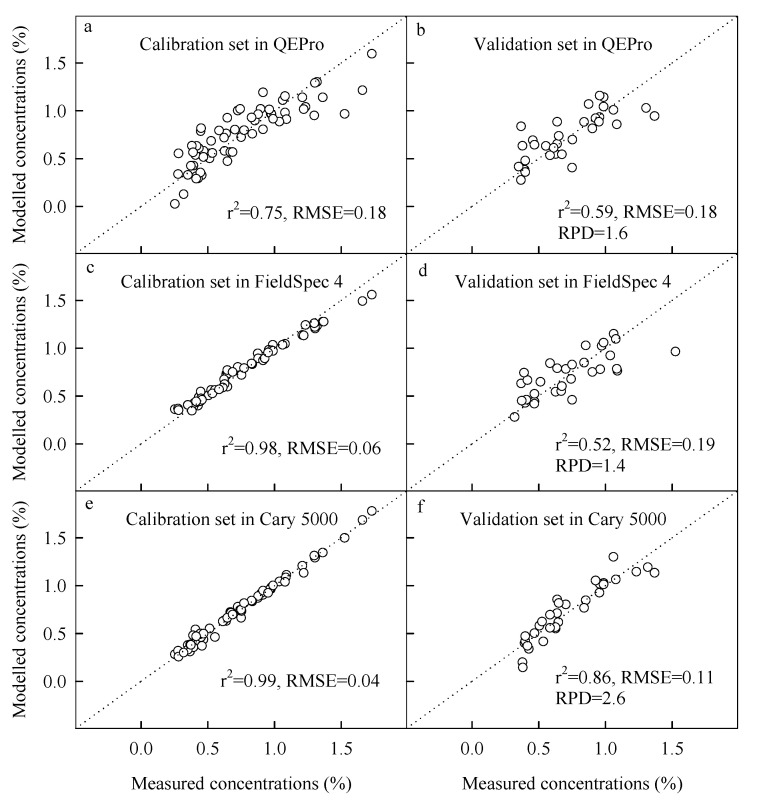
Results of spectral analysis on organic carbon concentrations in sediments of the Yellow Sea and Bohai Sea, China (spectral models for both the calibration set and validation set in QEPro (**a**,**b**), FieldSpec 4 (**c**,**d**), and Cary 5000 (**e**,**f**), respectively).

**Figure 6 sensors-22-05638-f006:**
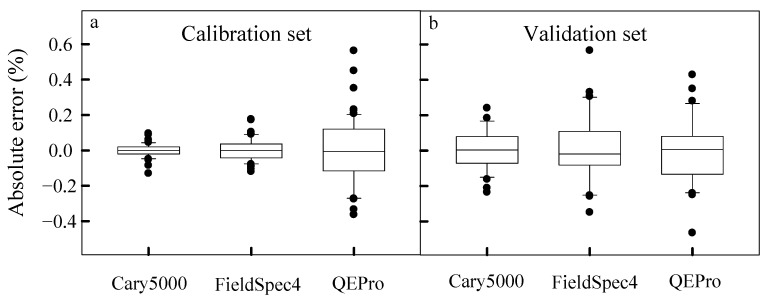
The quartile map of the absolute error for predicted organic carbon concentrations in the calibration set and validation set (absolute errors in the calibration set (**a**) and validation set (**b**)).

**Figure 7 sensors-22-05638-f007:**
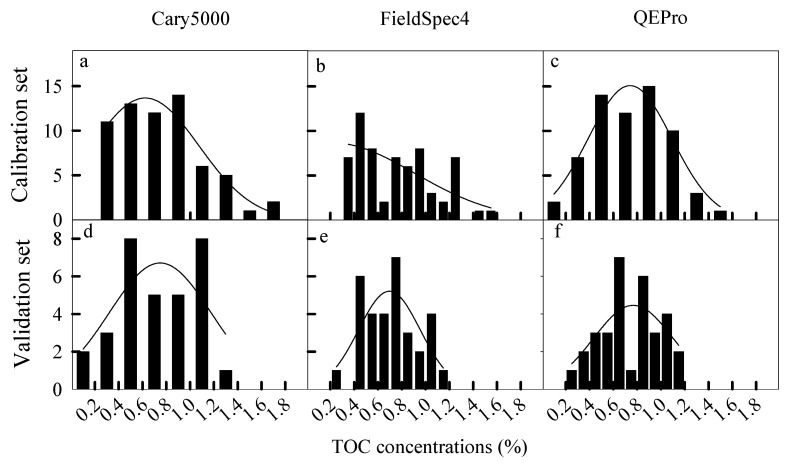
The distribution curves of predicted organic carbon concentrations by different spectral models established by both calibration set and validation set using different spectrometers (distribution curves from the calibration set by Cary 5000 (**a**), FieldSpec 4 (**b**), and QEPro (**c**), respectively; distribution curves from the validation set by Cary 5000 (**d**), FieldSpec 4 (**e**), and QEPro (**f**), respectively).

**Table 1 sensors-22-05638-t001:** Recent studies on spectral analysis of SOC by different spectrometers.

Unit	Spectrometers	Spectral Models in Validation Set	References
g·kg^−1^	ASD FieldSpec 4	R^2^ = 0.89, RMSE = 2.57, RPD = 3.04	[14]
	Agilent 4300	R^2^ = 0.98, RMSE = 1.12, RPD = 7.01	
g·kg^−1^	ASD FieldSpec 3	R^2^ = 0.96, RMSE = 2.1, RPD = 5.4	[15]
	Ocean Optics STS	R^2^ = 0.94, RMSE = 2.4, RPD = 3.9	
g·kg^−1^	ASD FieldSpec 3	R^2^ = 0.89, RMSE = 3.9, RPD = 2.9	[15]
	Ocean Optics STS	R^2^ = 0.85, RMSE = 4.2, RPD = 2.6	
%	Spectral Evolution FSR + 3500	R^2^ = 0.91, RMSE = 0.32, RPIQ = 1.31	[16]
	Spectral Engines OY MEMS S2.2	R^2^ = 0.80, RMSE = 0.46, RPIQ = 1.47	
g·kg^−1^	Silver Spring Foss XDS	RMSE = 0.23, RPIQ = 9.94	[17]
	Ettlingen Bruker-TENSOR	RMSE = 0.29, RPIQ = 8.01	
	ASD FieldSpec 3	RMSE = 0.83, RPIQ = 2.87	
	Agilent 4300	RMSE = 1.02, RPIQ = 2.28	
g·kg^−1^	East Norwalk	R^2^ = 0.54, RMSE = 4.1	[10]
	Ocean Optics USB2000 + Hamamatsu Photonics C9914GB	R^2^ = 0.49, RMSE = 4.5	
%	ASD AgriSpec	R^2^ = 0.89, RMSE = 0.12	[18]
	NeoSpectra	R^2^ = 0.78, RMSE = 0.16	
%	Bruker Optics Vertex 70	R^2^ = 0.96, RMSE = 0.17, RPIQ = 3.70	[19]
	Agilent 4200	R^2^ = 0.91, RMSE = 0.26, RPIQ = 2.46	
	ASD Labspec	R^2^ = 0.88, RMSE = 0.30, RPIQ = 2.13	

**Table 2 sensors-22-05638-t002:** The specifications of spectrometers used in this study.

Features	Cary 5000	FieldSpec4	QEPro
Sensor	Photodiode and TE cooled PbS	CCD (<1000 nm), InGaAs (>1000 nm)	Hamamatsu back–thinned FFT–CCD
Detector	Quartz window	Probe, fiber optic	Probe, fiber optic
Wavelength range	350–2500 nm	350–2500 nm	200–1100 nm
Optical resolution	1 nm	3 nm (700 nm) 10 nm (1400 nm–2100 nm)	0.3 nm
Signal-to-noise	>30,000	>10,000	1000
Integration time	100 ms	8 ms–15 min	100 ms
Stray light	<0.0002% (1420 nm)	0.02% (<1000 nm) 0.01% (>1000 nm)	<0.08% (600 nm) 0.4% (435 nm)
Wavelength repeatability	<0.02 nm (>750 nm) <0.005 nm (<750 nm)	0.1 nm	––
Wavelength accuracy	<0.4 nm (>750 nm) <0.08 nm (<750 nm)	0.5 nm	––

**Table 3 sensors-22-05638-t003:** The parameters to evaluate spectral models in both calibration set (C) and validation set (V), respectively.

Parameters	Cary 5000		FieldSpec 4		QEPro	
	C	V	C	V	C	V
Max	2.41	2.38	2.90	1.69	2.14	2.03
Min	0.01	0.00	0.00	0.00	0.01	0.00
Error	0.03	0.09	0.05	0.14	0.14	0.14
Resolution	0.006	0.005	0.005	0.004	0.006	0.004

## Data Availability

https://www.researchgate.net/publication/362290727_Reflectance_rates_and_orgainc_carbon_concentrations_of_the_sediments_in_the_Yellow_Sea_and_Bohai_Sea_China (accessed on 18 April 2022).

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
