# Peer review of "A Method to Evaluate Spectral Analysis by Spectroscopy"

_sensors, 2022, doi:10.3390/s22155638_

Round 1

Reviewer 1 Report

In this manuscript, the author reports, ‘A Method to Evaluate Spectral Analysis by Spectroscopy’. The authors should address the following questions before getting a possible publication.

Recommendation: Minor revisions needed as noted.

1.      The novelty of the present work should be discussed in the Introduction section.

2.    The formatting and grammatical errors in the article need to be checked carefully.

3.      The author should write the purpose for each test in one/two sentences (in brief) before explaining the results of the characterization techniques. Therefore, the logic and organization of this part will be enhanced.

4.      The authors have cited relevant references in the Introduction section; however the authors are encouraged to include some more recent references to broaden the impact.

Author Response

Response to Reviewer 1

Comments and Suggestions for Authors

In this manuscript, the author reports, ‘A Method to Evaluate Spectral Analysis by Spectroscopy’. The authors should address the following questions before getting a possible publication.

Recommendation: Minor revisions needed as noted.

Many thanks for the critical and valuable comments and suggestions.   We’ve carefully addressed the concerns raised by the reviewers:

  1. The novelty of the present work should be discussed in the Introduction section.

Response: Thanks for the critical and valuable suggestions and comments.  We have reorganized the introduction section and improved the discuss section, stressing the novelty of our work.

  1. The formatting and grammatical errors in the article need to be checked carefully.

Response: Thanks for the critical and valuable suggestions and comments.  We have carefully checked all the formatting and grammatical errors in the manuscript.

  1. The author should write the purpose for each test in one/two sentences (in brief) before explaining the results of the characterization techniques. Therefore, the logic and organization of this part will be enhanced.

Response: Thanks for the critical and valuable suggestions and comments.  We have improved the whole manuscript as the reviewer suggested.

  1. The authors have cited relevant references in the Introduction section; however, the authors are encouraged to include some more recent references to broaden the impact.

Response: Thanks for the critical and valuable suggestions and comments.  we have found and carefully read 7 papers highly related in our topics published in recent three years, and summarized a new table listed in the revision.  Then, we reorganized a new introduction in the revision. 

Reviewer 2 Report

I suggest minor revision for this paper and my comments are in below:   1-The English and grammar of the paper should be greatly improved in the revised version. 2-The authors should compare their work with at least five previously published works in the same area in one table before the conclusion section briefly. 3-More explanations regarding the Software name and its adjustments should be given in the revised version so that the future readers of the work can fallow it easier. 4-One paragraph in the introduction regarding to the applications of the method introduced by the authors should be presented.

Author Response

Response to Reviewer 2:

Many thanks for the critical and valuable comments and suggestions.  We’ve carefully addressed the concerns raised by the reviewers:

 I suggest minor revision for this paper and my comments are in below:  

  1. The English and grammar of the paper should be greatly improved in the revised version.

Response: Thanks for the critical and valuable suggestions and comments.  We improved the manuscript with the help of Dr. Zhao who works in the National Institute for Occupational Safety and Health, Spokane, WA, US ([email protected]).

  1. The authors should compare their work with at least five previously published works in the same area in one table before the conclusion section briefly.

Response: Thanks for the critical and valuable suggestions and comments.  We have found and carefully read 7 papers highly related in our topics published in recent three years, and summarized a new table listed in the revision.  Then, we reorganized a new introduction in the revision.

  1. More explanations regarding the Software name and its adjustments should be given in the revised version so that the future readers of the work can follow it easier.

Response: Thanks for the critical and valuable suggestions and comments.  we have performed the analysis in matlab 2017b and SPSS 13.0, a new program in R is preparing.  

  1. One paragraph in the introduction regarding to the applications of the method introduced by the authors should be presented.

Response: Thanks for the critical and valuable suggestions and comments.  We have reorganized a new introduction in the revision, in which we introduced how the method used in our work was inspired. 

Reviewer 3 Report

This work proposes and presents a data processing technique to validate spectroscopy analysis. The author uses four characteristics of spectrometer detection to predict sediment composition. The method is based on error estimation—the novelty of the work relies on the processing estimation and its simplicity. This technique can be attractive in some spectroscopy works. As a result, the manuscript will be suitable for publication in SENSORS after some minor points.

*The quality of Figure 1 needs an improvement. Please use English at the scale; it is hard to understand.

*The authors need to be clear about the terms ????? and RMSEC; they are confused in some parts of the text. Please also define these terms and their formula.

*The analysis of reflectance limit some spectroscopy applications. The authors need to include a discussion.

Author Response

Many thanks for the critical and valuable comments and suggestions.  We’ve carefully addressed the concerns raised by the reviewer:

*The quality of Figure 1 needs an improvement. Please use English at the scale; it is hard to understand.
Response: Thanks for the critical and valuable suggestions and comments.  We have deleted Figure 1 because it has been plotted in another paper [reference 24 in the new revision].

*The authors need to be clear about the terms ????? and RMSEC; they are confused in some parts of the text. Please also define these terms and their formula.
Response: Thanks for the critical and valuable suggestions and comments.  As the editor suggested, we have revised the confused term in the new revision and define its term and formula.

*The analysis of reflectance limits some spectroscopy applications. The authors need to include a discussion.

Response: Thanks for the critical and valuable suggestions and comments.  As the editor suggested, we have supplemented the limits in the discussion.

Reviewer 4 Report

The reviewed manuscript presents research on spectral detection of organic carbon in marine sediment samples from the Yellow Sea and Bohai Sea, China. These are my impressions and comments upon reading it:

1. It is not clear how it is possible to isolate absorption of organic carbon specifically from spectra of non-selective absorption within the ~500–2500-nm range. This should be explained thoroughly and clearly.

2. It is further not obvious how the 4 proposed parameters—detection range (max and min), resolution, and error—may attribute the data from non-selective absorption specifically to organic carbon. If this follows from the results of mathematical modelling, how is this possible to verify? How accurate is this attribution and to which extent does this method apply specifically to organic carbon or is this a general method?

3. Measurement of organic carbon concentration is not a new problem. This parameter is determined via multiple methods. The advantages of the proposed method need to be emphasised.

4. The Conclusion claims that the presented results corroborate the advantage of laboratory spectrometers over their portable counterparts. This ‘conclusion’, however, is patently obvious because laboratory spectrometers have better resolution, signal-to-noise ratio, &c and therefore they naturally cost much more. The Authors should remove or re-phrase this trivial comparison and give, instead, more prominence to the chosen portable spectrometer by providing precision and sensitivity that it delivers in measurement of organic carbon concentration.

If the Authors take the listed suggestions into consideration in a further revision of their paper, it may be published in Sensors

Author Response

Many thanks for the critical and valuable comments and suggestions.  We’ve carefully addressed the concerns raised by the reviewer:

  1. It is not clear how it is possible to isolate absorption of organic carbon specifically from spectra of non-selective absorption within the ~500–2500-nm range. This should be explained thoroughly and clearly.

Response: Thanks for the critical, valuable and professional comments.  In this work, we paid attention to the method for determining what the technical parameters such as detection range, resolution, and error were by visible and near infrared spectroscopy.  So, in this work we performed the modeling with the simplest spectral pretreatment, that is, we directly used the whole band information (non-selective spectra) to establish the relationship with sediment organic carbon concentrations.  Here, how to establish an accurate spectral model is not our end, which will be studied in our next step.

  1. It is further not obvious how the 4 proposed parameters—detection range (max and min), resolution, and error—may attribute the data from non-selective absorption specifically to organic carbon. If this follows from the results of mathematical modelling, how is this possible to verify? How accurate is this attribution and to which extent does this method apply specifically to organic carbon or is this a general method?

Response: Thanks for the critical, valuable and professional suggestions and comments.  

If we develop a technique or an equipment which could accurate and rapidly analyze SOC based on vis-NIR spectroscopy, we should present its technical parameters, in which the proposed 4 parameters are the most basic.  Therefore, the proposed 4 parameters are valuable for an equipment or technique based on vis-NIR spectroscopy.  

Now we don’t know how to verify the method by mathematics.  We’re poor in the theoretical mathematics, but we can make full use of the mathematical tools, such as SPSS, matlab, and R.  We know that the spectral model is highly depended on modeling samples, so we’re building the sample data.  We hope we could make the monitoring of soil or sediment more telligent by means of spectroscopy.  Welcome to joint with us to realize this great idea!

  1. Measurement of organic carbon concentration is not a new problem. This parameter is determined via multiple methods. The advantages of the proposed method need to be emphasised.

Response: Thanks for the critical, valuable and professional suggestions and comments.  And we can’t agree with the reviewer.  As the reviewer suggested, we stressed the advantage of the proposed method.

  1. The Conclusion claims that the presented results corroborate the advantage of laboratory spectrometers over their portable counterparts. This ‘conclusion’, however, is patently obvious because laboratory spectrometers have better resolution, signal-to-noise ratio, &c and therefore they naturally cost much more. The Authors should remove or re-phrase this trivial comparison and give, instead, more prominence to the chosen portable spectrometer by providing precision and sensitivity that it delivers in measurement of organic carbon concentration.

Response: Thanks for the critical, valuable and professional suggestions and comments.  And we can’t agree with the reviewer.  As the reviewer suggested, we revised the related parts in discussion.

If the Authors take the listed suggestions into consideration in a further revision of their paper, it may be published in Sensors

Response: Many thanks for the critical, valuable and professional suggestions and comments.  We’ve carefully addressed the concerns raised by the reviewers.  We hope this revision is now appropriate for the publication by Sensors.

Reviewer 5 Report

1.         The paper compared three commercial products such as Agilent Cary 5000, ASD FieldSpec 4 and Ocean Optics QEPro. This is not a scientific paper, but a laboratory note. Therefore, the paper does not match the journal scope.

2.         The authors claimed that “In the field of visible and near infrared spectroscopy (Vis-NIRS), studies have concentrated on how to establish a good model, but ignored how to evaluate the spectral analysis”. This is completely wrong. Spectral analysis is the main issue of spectroscopic studies.

3.         Please use a professional English editing service.

Author Response

  1. The paper compared three commercial products such as Agilent Cary 5000, ASD FieldSpec 4 and Ocean Optics QEPro. This is not a scientific paper, but a laboratory note. Therefore, the paper does not match the journal scope.

Response: Many thanks for the critical comments and suggestions.  We’ve carefully addressed the concerns raised by the reviewer.  We found in the old manuscript, we have not told the logic and interesting story.  Therefore, in the new revision, we reorganized a new introduction and also improve the whole story.  We hope this revision is now appropriate for the publication by Sensors.

  1. The authors claimed that “In the field of visible and near infrared spectroscopy (Vis-NIRS), studies have concentrated on how to establish a good model, but ignored how to evaluate the spectral analysis”. This is completely wrong. Spectral analysis is the main issue of spectroscopic studies.

Response: Many thanks for the critical comments and suggestions.  We’ve carefully addressed the concerns raised by the reviewer.  We hope we could make the monitoring of soil or sediment more telligent by means of spectroscopy.  Therefore, spectral analysis is the main pressure which was used to support for developing a technique or an equipment which could accurate and rapidly analyze SOC based on vis-NIR spectroscopy.

  1. Please use a professional English editing service.

Response: Many thanks for the critical comments and suggestions.  We’ve published many scientific papers in English, but our English also still needs to be improved.  So, we improved the manuscript with the help of Dr. Zhao who works in the National Institute for Occupational Safety and Health, Spokane, WA, US ([email protected]). 

Round 2

Reviewer 4 Report

The revised manuscript brings a good deal of changes, but it is not clear which of them have been made in order to address my questions and concerns. Furthermore, the new version does not in general clarify the circumstances that gave rise to the majority of my questions. It is necessary to specify, which parts of the new revision were added in answer to the reviewer’s questions

Author Response

Many thanks for the critical and valuable comments and suggestions.  We’ve carefully addressed the concerns raised by the reviewer:

  1. It is not clear how it is possible to isolate absorption of organic carbon specifically from spectra of non-selective absorption within the ~500–2500-nm range. This should be explained thoroughly and clearly.

Response: Thanks for the critical, valuable and professional comments.  In the second revision, we explained why we used the whole spectra without characteristic spectra extraction to establish spectral models in 2.3 Spectral analysis.  All changes were marked in red font.  

  1. It is further not obvious how the 4 proposed parameters—detection range (max and min), resolution, and error—may attribute the data from non-selective absorption specifically to organic carbon. If this follows from the results of mathematical modelling, how is this possible to verify? How accurate is this attribution and to which extent does this method apply specifically to organic carbon or is this a general method?

Response: Thanks for the critical, valuable and professional suggestions and comments. 

The proposed parameters were derived from spectral models of organic carbon.  And whether spectral models were good highly depended on the quality of modeling samples and spectrometers.  Therefore, the proposed parameters were highly affected by the quality of modeling samples and spectrometers.  In this study, we only verified the influence of spectrometers on these parameters, but had no more other samples to verify how samples influenced the proposed parameters.  

And this method is general for all spectral analysis. We have added the specific procedure in SPSS 13.0.  All changes were marked in red font.

  1. Measurement of organic carbon concentration is not a new problem. This parameter is determined via multiple methods. The advantages of the proposed method need to be emphasized.

Response: Thanks for the critical, valuable and professional suggestions and comments.  And we can’t agree with the reviewer.  As the reviewer suggested, we stressed the advantage of the proposed method.  All changes were marked in red font.

  1. The Conclusion claims that the presented results corroborate the advantage of laboratory spectrometers over their portable counterparts. This ‘conclusion’, however, is patently obvious because laboratory spectrometers have better resolution, signal-to-noise ratio, &c and therefore they naturally cost much more. The Authors should remove or re-phrase this trivial comparison and give, instead, more prominence to the chosen portable spectrometer by providing precision and sensitivity that it delivers in measurement of organic carbon concentration.

Response: Thanks for the critical, valuable and professional suggestions and comments.  And we can’t agree with the reviewer.  As the reviewer suggested, we revised the related parts in discussion.  All changes were marked in red font.

If the Authors take the listed suggestions into consideration in a further revision of their paper, it may be published in Sensors

Response: Many thanks for the critical, valuable and professional suggestions and comments.  We’ve carefully addressed the concerns raised by the reviewers.  We hope this revision is now appropriate for the publication by Sensors.

Reviewer 5 Report

Thank you for your response. The manuscript was drastically improved.

Author Response

Many thanks for the critical and valuable review!

Round 3

Reviewer 4 Report

The manuscript under review studies a method of fast detection of organic carbon in soils or sediments. I am not satisfied with the way the Authors answered my concerns. First of all, their answers were provided in private messages and it is not clear whether or not my comments were addressed in the next revision of the manuscript (most likely, not). Secondly, the Authors’ responses did not put my questions to rest: the parts of the manuscript concerned are still as vague as before. It is necessary to provide responses to my comments directly in the text of the manuscript, and those responses should be thorough. This manuscript needs further revision and may only be published after detailed responses to my questions are given in its text

Author Response

Many thanks for the critical and valuable suggestions and comments.  We’ve carefully and directly addressed the concerns raised by the reviewer in the new revision and listed a rebuttal against each point in the cover letter.  We hope this revision is now appropriate for the publication by Sensors.

  1. It is not clear how it is possible to isolate absorption of organic carbon specifically from spectra of non-selective absorption within the ~500–2500-nm range. This should be explained thoroughly and clearly.

Response: Thanks for the critical and valuable comments.  In the second revision, we explained why we used the whole spectra without characteristic spectra extraction to establish spectral models in 2.3 Spectral analysis.  All changes were marked in red font.  

  1. It is further not obvious how the 4 proposed parameters—detection range (max and min), resolution, and error—may attribute the data from non-selective absorption specifically to organic carbon. If this follows from the results of mathematical modelling, how is this possible to verify? How accurate is this attribution and to which extent does this method apply specifically to organic carbon or is this a general method?

Response: Thanks for the critical and valuable suggestions and comments.  

The proposed parameters were derived from spectral models of organic carbon.  And whether spectral models were good highly depended on the quality of modeling samples and spectrometers.  Therefore, the proposed parameters were highly affected by the quality of modeling samples and spectrometers.  In this study, we only verified the influence of spectrometers on these parameters, but had no more other samples to verify how samples influenced the proposed parameters.  

And this method is general for all spectral analysis. We have added the specific procedure in SPSS 13.0.  All changes were marked in red font.

  1. Measurement of organic carbon concentration is not a new problem. This parameter is determined via multiple methods. The advantages of the proposed method need to be emphasized.

Response: Thanks for the critical and valuable suggestions and comments.  And we can’t agree with the reviewer.  As the reviewer suggested, we stressed the advantage of the proposed method.  All changes were marked in red font.

  1. The Conclusion claims that the presented results corroborate the advantage of laboratory spectrometers over their portable counterparts. This ‘conclusion’, however, is patently obvious because laboratory spectrometers have better resolution, signal-to-noise ratio, &c and therefore they naturally cost much more. The Authors should remove or re-phrase this trivial comparison and give, instead, more prominence to the chosen portable spectrometer by providing precision and sensitivity that it delivers in measurement of organic carbon concentration.

Response: Thanks for the critical and valuable suggestions and comments.  And we can’t agree with the reviewer.  As the reviewer suggested, we revised the related parts in discussion.  All changes were marked in red font.

This manuscript is a resubmission of an earlier submission. The following is a list of the peer review reports and author responses from that submission.